# OpenReview forum: "Robust Reward Modeling via Causal Rubrics"
_ICLR.cc/2026/Conference — ICLR 2026 Poster_

### Official Review · Reviewer_Ge7h · 2025-10-30

**Soundness:** 3
**Presentation:** 4
**Contribution:** 3
**Rating:** 6
**Confidence:** 4

**Summary:**

The paper introduces “causally robust reward modeling” (CROME), a data augmentation technique for training reward models for RLHF-based language model alignment that encourages them to focus on meaningful attributes that differentiate good from bad answers, rather than spurious ones (such as response length or specific formatting patterns).  The approach uses a teacher LM to generate, for a given preference example in conventional reward model training datasets, an oracle LM-produced rubric that explicitly lists meaningful (“causal”) attributes, then synthesizes alternative answers that are supposed to alter only the meaningful attributes (from both the preferred and dispreferred answers in the original preference data).  They also generate alternative questions to which both answers should receive similar reward values, aiming to help the reward model learn to ignore spurious attributes.  Relative to existing methods that attempt to improve reward models by data augmentation, this work claims to be superior in avoiding the need to explicitly state what the spurious attributes are, exactly.  The paper includes a bit of theory to explain why, under the causal assumptions made by the framework, the approach should work.  Extensive experiments show that it does, in fact, lead to better rewards and better aligned language models, on a range of benchmarks spanning safety, reasoning, and general instruction following.

**Strengths:**

The core idea is elegant and generally clearly explained.

The ideas behind Crome and the specific setup – augmenting data while focusing on causal/spurious attributes, appear rather novel and well motivated.

The experiments are extensive and present strong empirical evidence that the approach is more effective than the baselines selected, across a range of settings and diverse benchmarks.  The paper is quite thorough in using extant base models and benchmarks to evaluate whether the approach performs as intended, and extensive ablations, such as varying the oracle LM. It is convincing that the advantages provided by Crome are robust.

**Weaknesses:**

The main paper does not include a single example.  The formal notation is appreciated, but the intuition would be a lot easier to grasp with an accompanying example.  (E.g., I got hung up on the phrase “flipping the question” in one of the figures.  Awkward nomenclature would matter a lot less if the reader saw a simple illustration of the idea.)

Crome relies heavily on the oracle LM. The central assumption that language models can generate reliable causal rubrics is worth evaluating empirically. Even though the empirical results of training with the rubrics are impressive, there is no attempt to confirm the quality of the rubrics generated by the teacher LM, or of the synthesized examples, even informally.  In other words the “approximations” of line 208 are accepted without any attempt to quantify how good they are.

Stemming from the previous point, the biases of the oracle might propagate into the rubric and into the final augmented training data. For example, if the oracle strongly prefers longer responses, then “longer responses” might be identified as a causal attribute in the rubric, and Crome does not offer a robust method to guard against that.

There is no discussion of exactly how much data augmentation is done (how many of each kind of example) or the cost of doing so (either to train the RM or inference cost in producing the new examples).  The proposed framework (augmenting + filtering per example) seems rather expensive (the method requires generating 10+1=11x data initially and then filtering).  This should be quantified so readers understand the cost of the approach, especially as datasets grow.  Ideally we’d also see some analysis of the cost-benefit tradeoff (e.g., what happens to the performance gains if I do half as much data augmentation?).

Minor suggestions:
Figure 6 labels need to be explained somewhere in the caption or paper text.

**Questions:**

Based on my understanding Crome divides attributes into two binary categories – spurious and causal. Is it possible to generalize this idea, accounting for qualitative differences in the level of importance of different attributes, or capturing the ways in which different attributes interact?

How consistent are the rubrics produced by the oracles across different runs or different choices of oracles? Are similar attributes consistently classified similarly? I would appreciate more quantitative experiments specifically studying the rubrics and the attributes identified between oracles.

---

> ### Author Response · Authors · 2025-11-21
> **Author Response - 1**
>
> Thank you for the positive and encouraging feedback highlighting the clarity, novelty, and robustness of CROME, as well as the strength and thoroughness of our experimental evaluation. Please see our responses below.
>
>
> >**Quality of rubrics and human evaluation**
>
>
> Before initiating large-scale data generation, we indeed conducted a preliminary human evaluation study to measure the relevance of rubrics, as well as qualitative analysis of multiple examples to assess rubric relevance. This served as inspiration for large-scale and question-specific rubric generation. In particular, we conducted a small-scale human study across 112 prompts using a set of 60 rubrics derived from MT-Bench using the same method of causal rubrics extraction. 2 human annotators established the ground-truth relvant rubrics (from this set) for each prompt. We then measured the alignment of the Gemini 2.0 Flash model's top-5 ranked rubrics against this human gold standard for every prompt.
> Our results demonstrate strong alignment:
> - **High Retrieval Precision (MRR)**: The mean reciprocal rank achieved was 0.9359. This confirms the model is highly effective at placing at least one human-validated relevant rubric at the very top rank.
> - **High Ranking Quality (nDCG@5)**: The average nDCG (Normalized Discounted Cumulative Gain) at K=5 was 0.7795. Since nDCG uses the human-annotated relevance as its gain, this score confirms that the model consistently ranks the attributes humans deem most relevant highly.
> This empirical alignment validates that the rubrics ranked by LLMs have strong alignment with humans. While the human study was performed using a fixed library of rubrics, for large-scale rubric generation and counterfactual data augmentation, we remove the constraint of a fixed library and allowed free-form rubric generation (constrained by our guidelines presented in the prompt to the oracle).
>
> **See Author Response - 3 below for results on consistency of rubrics across oracle choices**
>
> >**Examples and Illustrations in the main paper**
>
>
> Thanks for suggesting. We have included Table 2 in the maine text of the updated paper adding brief examples of causal and neutral augmentations for better clarity and grasp of the method. Additionally appendix Section J has a detailed walkthrough with more details.

---

> ### Author Response · Authors · 2025-11-21
> **Author Response - 2**
>
> >**Discussion on the amount and cost of data augmentation and cost-benefit tradeoff.**
>
> Thanks for this important suggestion. Appendix Section C.5 and F.4 includes this information. We updated main paper Section 6.2 (page 10) with cost-matched experiments as well (with appropriate references), and provide these important details below.
>
> **Amount of data augmentation:**
>
> We generate 5 causal attributes per question, and for each, the oracle generates a causally upgraded and degraded attribute, resulting in 10x the dataset size, and an equal amount of Neutral augmentations. Augmented data and original data is combined resulting in 21x the dara size. After verification and filtering, we get 3.5x the data size, similar to RRM. Below, we mention the cost of data augmentation, training RRM and CROME, as well as provide cost-matched experiments.
>
> **On Quantifying the cost:**
>
> The cost of inference for our runs is around 50% of the full training cost, as shown below:
>
> - Training cost of RRM is 15 hours of compute, 8×A100s. 20 USD/hr, total training cost is: 20 USD/hr × 15 hr = 300 USD for a standard GCP instance.
> - Inference cost for augmentations for 600k responses at  0.4USD/M output token cost (for Gemini Flash API) costs about 120 USD. This is conservatively < 50% of the training cost of RRM.
>
> **Budget-Matched Experiment:**
>
> We conducted a new budget-matched experiment. We gave the RRM baseline an additional 25%, and 50% of standard preference data, matching CROME's augmentation budget. This could be done since RRM uses only half their augmented data in their recipe. We also experiment with reducing the amount of data for CROME to match the cost. The results below show that this data-boosted RRM still significantly underperformed CROME. This confirms that CROME's structured, causally-guided augmentations are more sample-efficient than simply adding more preference pairs.
> Rewardbench and ReWordBench results:
>
>
> | Model | #Examples | Chat | ChatHard | Safety | Reasoning | Average RewardBench | Average ReWordBench |
> |-------|-----------|------|-----------|---------|-----------|------------------|------------------|
> | RRM   | X×1.5     | 97.63 | 71.16 | 74.26 | 87.13 | 82.55 | 64.53 |
> | RRM   | X×1.25    | 97.63 | 71.71 | 74.59 | 87.10 | 82.76 | 64.54 |
> | RRM   | X         | 96.93 | 72.04 | 73.78 | 87.36 | 82.53 | 63.92 |
> | **CROME** | **X** | **97.49** | **72.70** | **86.96** | **94.55** | **87.93** | **73.07** |
>
> *Here, X = # original RRM data ,approximately 230k examples*
>
> On ReWordBench, CROME is better than RRM on 21, 20 and 20 out of 23 transformations, for X×1.5, X×1.25 and X number of Examples, respectively.
>
> Additionally, we find that when we train CROME with lesser data, equal to X/1.5 and X/1.25 amounts, we get average RewardBench accuracy to be **85.95** and **85.81** respectively, higher than the original RRM score of 82.53, and average ReWordBench accuracy to be **73.66** and **73.51** respectively, significantly higher than original RRM score of **63.92**, and not much different from CROME trained on X amount of data.
>
>
>
> >**Minor suggestions: Figure 6 labels need to be explained somewhere in the caption or paper text.**
>
>
> Thank you for the suggestion. We explain these labels in Appendix Section C.8, and as per your suggestion, we have now edited Section 6.2 to have a brief explanation of the labels in the main text as well.
>
> >**Based on my understanding, CROME divides attributes into two binary categories – spurious and causal. Is it possible to generalize this idea, accounting for qualitative differences in the level of importance of different attributes, or capturing the ways in which different attributes interact?**
>
>
>
> As the reviewer correctly mentions, currently in our theoretical analysis (Section 5), we assume a sparse model of dependence of final learned reward to a subset of attributes (both causal and spurious) and show that training on data from targeted, ideal counterfactual interventions enables the model to identify the true causal attributes that determine the reward.
> - Qualitative differences in the level of importance of different attributes: It would be interesting to measure if LLMs can reliably provide level of importance of different causal attributes as well, which would allow using these importance signals in loss formulation or data selection for traiing (post counterfactual augmentation), making it an important future direction.
> - Capturing ways in which attributes interact: We do take into account correlation between spurious and causal variables, and design irrelevant query neutrals that counteract the effect of spurious variables being correlated with casuals.

---

> ### Author Response · Authors · 2025-11-21
> **Author Response - 3**
>
> >**Consistency of rubrics across oracle choices**
>
>
>
> **We analyse the similarity of rubrics obtained using different oracle LLMs, namely Gemma-3-27B-IT and Gemini-2.0-Flash**. We perform this study on 50 random examples from the ultrafeedback dataset. Qualitatively we see high overlap (>70%) between rubrics from the 2 oracles. For quantitative results, we prompt SoTA LMs - gpt-5.1-2025-11-13 and gemini-3-pro-preview to obtain the set of overlapping elements between 2 sets of rubrics identified by Gemma-3-27B-IT and Gemini-2.0-Flash, for the 50 examples from the ultrafeedback data. We find that on average, the number of overlapping rubrics is greater than 3.72 out of 5, or 74.4\% (using gpt-5.1-2025-11-13) and 3.52 out of 5 or 70.4\% (using gemini-3-pro-preview).
> See Appendix Section N (second last page, page 58 of the paper) for details and Table 16 (last page) where few examples from ultrafeedback rubrics obtained from Gemma-3-27B-IT and Gemini-2.0-Flash, along with overlapping rubrics are provided.

---

> > ### Author Response · Authors · 2025-11-28
> > **Gentle Follow Up**
> >
> > We wanted to follow up and thank you for your time and insightful feedback.
> >
> > **Your suggestions have helped us greatly improve our paper, including:**
> > 1. **Discussion on the amount and cost of data augmentation and budget-matched experiments (Appendix Section C.5)**
> > 2. **Addition of examples and illustrations as part of the main paper (Table 2 of the main paper)**
> > 3. **Analyzing consistency of rubrics across oracles (Section N of main paper)**
> > 4. **Estimation of quality of rubrics via human evaluation.**
> >
> > ----
> >
> > *We have summarized the new results, analyses, and updates to the paper as part of the general response.*
> >
> > Should these responses address your concerns, we would greatly appreciate this being reflected through an increase in score. Thank you again for your engagement throughout this process.

---

### Official Review · Reviewer_679r · 2025-10-31

**Soundness:** 3
**Presentation:** 2
**Contribution:** 3
**Rating:** 8
**Confidence:** 4

**Summary:**

This paper uses a large (potentially proprietary) language model to create rubrics for prompts, and to perturb potential responses to that prompt, to design a dataset to train a more robust reward model. They try to mitigate two failure cases in reward models: when a RM does not always correctly identify which response is worse (due to potentially small changes in the response), and when a RM is biased towards a specific response for factors that do not have any actual bearing on the quality of the response. They show strong gains on popular reward modeling benchmarks.

**Strengths:**

This paper's experiments are carefully designed, and focused on an active area of work with reasonable baselines and good results on popular benchmarks. Their work defines clear methodology for using synthetic data generation methods to create augmentations to preference data, creating more robust and effective reward models. They also evaluate their results on best-of-n rankings for popular benchmarks, which has shown to be even better correlated with downstream performance after performing online RL than traditional RM benchmarks.

**Weaknesses:**

Some of the mathematical notation feels unnecessary, and I feel like it obfuscates the (reasonable) points being made at times. E.g. much of the mathematical description in sections 3 and 4 could instead be turned into natural prose, which would be more readily understandable to people less familiar with reward modeling, etc. The points your paper is making are good, but it can be hard to fully parse the paper at times.

**Questions:**

I'd recommend evaluating on the newer version of RewardBench: https://arxiv.org/abs/2506.01937, which includes a metric measuring how well models rate "tied" responses, as this would be quite relevant to your setup!

---

> ### Author Response · Authors · 2025-11-21
> **Author Response**
>
> Thank you for the encouraging feedback on our experimental design and for recognising the clear methodology for using synthetic data generation to create more robust and effective reward models.
>
> >**Scores on RewardBench2 and performance on the ties subset.**
>
>
> Thank you for the great suggestion. We now evaluate CROME and baselines on RewardBench2, as shown in the table below. We find that CROME outperforms RRM and RM on the average score (overall) by approximately 1.5\% and 5.5\%, respectively, and improves on the TIES subset by approximately 2\% and 4\% over RRM and RM, respectively, signifying improved calibration. See the full results in the table below:
>
> | Model | Overall | Factuality | Precise IF | Math | Safety | Focus | Ties (Accuracy) | Ties (Reward Margin Calibration) | Ties (Weighted Avg) |
> |-------|---------|------------|------------|------|---------|--------|------------------|---------------------|---------------------|
> | RM    | 57.97 | 58.74 | 33.75 | 63.93 | 54.89 | 58.59 | 84.31 | 71.57 | 77.94 |
> | RRM   | 61.97 | 55.79 | 37.50 | 66.12 | 57.33 | 75.15 | 86.27 | 73.53 | 79.90 |
> | CROME | 63.56 | 65.26 | 31.87 | 61.75 | 66.67 | 73.94 | 87.25 | 76.47 | 81.86 |
>
> Note that Ties (Weighted Avg) is the official metric from the rewardbench2 paper which is a weighted average between Ties (Calibration) and Ties (Accuracy) reported above. Ties (Accuracy) measures the rate at which all the chosen responses are ranked higher than all the rejected responses, while Ties (Reward Margin Calibration) score measures if the score margins between the lowest-ranked chosen and the highest-ranked rejected is larger than those between the highest and lowest-ranked chosen responses. CRoME shows approx. 3% boost in calibration showing that CRoME is not just accurate, it is well-calibrated in its preference ranking and doesn’t exhibit any arbitrary preference bias. It understands that a correct-vs-incorrect distinction is much more important than a correct-vs-correct distinction. This translates to a more robust and less arbitrary reward signal for downstream policy training and test-time scaling.
>
>
>
> >**Mathematical notation and clarity of the paper**
>
>
> Thank you for important feedback for improving the clarity of the paper. We are working on reducing notations for sections other than Section 5 (theoretical analysis).

---

> ### Author Response · Authors · 2025-11-28
> **Gentle Follow Up**
>
> We wanted to follow up and thank you for your time and insightful feedback on evaluating RewardBench 2 (and the TIES subset), which further strengthened our work by showing that CROME is well-calibrated in its preference ranking.
>
> As per your suggestions, we have improved the clarity of the paper by revisiting Sections 3 and 4 of the main paper and adding more natural prose and explanations alongside the mathematical notation to improve readability, with the heavier causal notations moving to the appendix.
>
> ---
>
> *Please see Sections 3 and 4 of the main paper and the General Response above for a summary of the changes made to improve the clarity of the paper. Additionally, we have summarized all new results, analyses, and updates to the paper as part of the general response.*

---

### Official Review · Reviewer_Hbwu · 2025-11-01

**Soundness:** 3
**Presentation:** 2
**Contribution:** 3
**Rating:** 2
**Confidence:** 4

**Summary:**

The paper proposes a method to generate synthetic preference data for more robust preference reward modeling. The method extracts “causal” attributes from preference pairs using LLMs, and given these verbalized rubrics counterfactually generates chosen and rejected pairs. The dataset is further augmented with ties pairs. The goal is to reduce the reward models’ reliance on spurious correlations. For their experiments they train either a RM or a DPO model on that data and evaluate it on preference benchmarks and Best-of-N setups. Compared to baselines, they show improvements on most of the evaluations, especially in the safety domain.

**Strengths:**

The paper tackles the important problem of improving preference reward models’ robustness and sensitivity towards spurious correlations. Their synthetic data generation approach both addresses how aware reward models are of real/important attributes in the completions vs. how invariant they are towards spurious attributes.
They performed a large set of experiments to showcase the effectiveness of their approach, such as evaluating their model on RewardBench and reWordBench, which is more adversarial. They also show how Crome (their approach) is less easily distracted by spurious correlations in the data, such as length or disguised safety instances.

**Weaknesses:**

- Clarity: The paper’s writing could be improved and certain experimental design choices could be explained more clearly. Additionally, the paper’s main text very often discusses results from tables in the appendix, which makes it harder to read. Some examples where clarity could be improved are: It is never specified what reward model is being trained, at what size (same for the baselines).
- Analysis of the data: The paper is missing some details on the synthetic data that you generate, such as a quantitative and qualitative analysis.
- Terms are not used precisely: I worry that the terms “causal” and “counterfactual” might be slightly overselling the approach.
- Baselines: As the baselines are barely explained in the paper, I wonder whether they are weak and whether you compared the approach to other current SOTA reward models.
- It would be good to see scores on RewardBench2, which is more challenging and explicitly contains a ties evaluation. Since the Crome approach also trains on ties data, it would be good to see performance on the ties subset.

**Questions:**

- Do you see any benefit in still incorporating more “subjective” preferences in preference tuning? While there might be spurious correlations, certain stylistic preferences could still provide relevant signal for improving an LLMs output distribution


- This approach will probably not prevent reward hacking completely? How do you test for that?

- 90: are the causal attributes specified during training?

- 174: don’t understand: what does it mean for a model to be conceptual?

- 204: how do you check these generations?
- 240: how do you make sure that the irrelevant query is indeed irrelevant?
- 270: did you do ablations on whether this is actually necessary?

- Section 6.1: you never mention the size of the RMs that you train, or what they are

- Table 2: main improvements in safety? Why?


- 375: baselines are not explained at all

- Table 5: unclear what is being compared, i.e. what are all the models?

- 460: effect of causal attributes: but should we completely ignore things like length etc.?

- 465: is that testing whether crome outperforms llm-as-judge?

- I could need more detail on data + how much you generate etc.

---

> ### Author Response · Authors · 2025-11-21
> **Author Response - 1**
>
> Thank you for the positive feedback on the importance of the robustness problem being addressed, recognizing the merits of our synthetic data generation approach, and highlighting the strength of our experimental evaluation.
>
> >**Scores on RewardBench2 and performance on the ties subset.**
>
>
> Thank you for the great suggestion. We now evaluate CROME and baselines on RewardBench2, as shown in the table below. We find that CROME outperforms RRM and RM on the average score (overall) by approximately 1.5\% and 5.5\%, respectively, and improves on the TIES subset by approximately 2\% and 4\% over RRM and RM, respectively, signifying improved calibration. See the full results in the table below:
>
> | Model | Overall | Factuality | Precise IF | Math | Safety | Focus | Ties (Accuracy) | Ties (Reward Margin Calibration) | Ties (Weighted Avg) |
> |-------|---------|------------|------------|------|---------|--------|------------------|---------------------|---------------------|
> | RM    | 57.97 | 58.74 | 33.75 | 63.93 | 54.89 | 58.59 | 84.31 | 71.57 | 77.94 |
> | RRM   | 61.97 | 55.79 | 37.50 | 66.12 | 57.33 | 75.15 | 86.27 | 73.53 | 79.90 |
> | CROME | 63.56 | 65.26 | 31.87 | 61.75 | 66.67 | 73.94 | 87.25 | 76.47 | 81.86 |
>
> Note that Ties (Weighted Avg) is the official metric from the rewardbench2 paper which is a weighted average between Ties (Calibration) and Ties (Accuracy) reported above. Ties (Accuracy) measures the rate at which all the chosen responses are ranked higher than all the rejected responses, while Ties (Reward Margin Calibration) score measures if the score margins between the lowest-ranked chosen and the highest-ranked rejected is larger than those between the highest and lowest-ranked chosen responses. CRoME shows approx. 3% boost in calibration showing that CRoME is not just accurate, it is well-calibrated in its preference ranking and doesn’t exhibit any arbitrary preference bias. It understands that a correct-vs-incorrect distinction is much more important than a correct-vs-correct distinction. This translates to a more robust and less arbitrary reward signal for downstream policy training and test-time scaling.

---

> > ### Comment · Reviewer_Hbwu · 2025-11-24
> >
> > Thank you for showing new results on the Ties subset of RewardBench 2. It's good to see that your approach considerably outperforms RRM and RM, both in terms of calibration and accuracy! I have one more question about this evaluation: I see that there models in the RewardBench 2 paper (Table 3), that achieve higher scores on the ties subset. Could you explain why CROME does not outperform them, or why you don't compare against those reward models?

---

> ### Author Response · Authors · 2025-11-21
> **Author Response - 2**
>
> >**Analysis of the data: The paper is missing some details on the synthetic data that you generate, such as a quantitative and qualitative analysis.**
>
>
> We appreciate the reviewer's request for a detailed analysis of the synthetic data generated by the CRoME framework. We agree that rigorous quantitative and qualitative analysis is essential to ensure our synthetic preference data represents meaningful alignment shifts and does not introduce new spurious correlations.
>
> To address this, we conducted an analysis focusing on three popular spurious correlations: structural bias (length), manipulative or sycophantic behaviors, and safety adherence (refusals). For obtaining sycophancy scores and the refusal rates on safety-critical queries, we use a prompted GPT-4.1-mini Oracle. The Oracle assigns a score on a 5-point Likert scale where 1 stands for least sycophancy and 5 stands for extreme sycophancy We ran the above evaluation 3 times with each run consisting of 100 randomly sampled queries. In case of sycophancy, we prompt the gpt4.1-mini to gauge the extent to which the response reinforces user beliefs and does excessive flattery. For safety, we ask the model to first identify if the query is malicious (safety critical) and then classify whether the response is a refusal without any leakage of dangerous information.
>
> - Does the CRoME intervention introduce or exploit structural biases, specifically response length, when generating new preference pairs?
> - Does improving a response along a specific dimension (e.g.,Instruction-following) inadvertently increase the response's sycophancy score?
> - Can the CRoME framework be effectively used to precisely shift the model's safety boundary, specifically by increasing the likelihood of a correct refusal to a safety-critical prompt?
>
> | Metric                             | Group                    | Mean Value | Std Dev |
> | ---------------------------------- | ------------------------ | ---------- | ------- |
> | Length (Number of Tokens)          | Base Chosen (Original)   | 309.55     | 251.87  |
> |                                    | Base Rejected (Original) | 231.19     | 220.27  |
> |                                    | CRoME Upgraded Response  | 329.86     | 380.98  |
> |                                    | CRoME Degraded Response  | 208.76     | 209.4   |
> |                                    |                          |            |         |
> | Sycophancy Score [1-5]                  | Base Rejected (Original) | 1.15       | 0.001   |
> |                                    | CRoME Upgraded Response  | 1.13       | 0.05    |
> |                                    |                          |            |         |
> | Refusal Rate (% on Safety Prompts) | Base Rejected (Original) | 17.22%     | 9.48%  |
> |                                    | CRoME Upgraded Response  | 21.33%     | 13.28%  |
>
> We find that although the mean number of tokens are higher in the upgraded versions of the originally rejected responses, it doesn’t cause a bias towards lengthier responses as shown in our Table 7 on  Length (Verbosity) Bias Analysis and CRoME doesn’t reward responses which are verbose but do not adhere to the contextual rubrics.  Legitimate Increase in refusal rates for safety critical queries also explains the gains in safety.
>
> Please let us know if you feel any other analysis of the data would further help your evaluation and the clarity of the paper.
>
>
> >**I worry that the terms “causal” and “counterfactual” might be slightly overselling the approach.**
>
> Please refer to section 3.3 in the main paper, where we explicitly explain that an ideal counterfactual is obtained when only one causal attribute is changed and all other exogenous factors (that produced the factual answer) remain the same. However, generating such ideal textual counterfactuals is intractable; hence, CROME employs Large Language
> Models (LLMs) produce approximate counterfactuals by rewriting answers, and while these are imperfect, they provide the targeted variations crucial for our data augmentation.
>
> By being upfront about our contributions, which include providing a spurious-unaware causal model and its practical realization through the use of LLMs, as mentioned in the paper, we hope to avoid being perceived as overselling our contributions. We would be happy to make changes in presentation as the reviewer suggests.

---

> > ### Comment · Reviewer_Hbwu · 2025-11-24
> >
> > Thank you for the data analysis, this is great! I am also satisfied with your answer regarding terminology, subjective preferences, and irrelevant queries!
> >
> > Regarding baselines: Could you clarify what data (and how many data instances) each baseline was trained on?

---

> ### Author Response · Authors · 2025-11-21
> **Author Response - 3**
>
> >**Details of Baselines and comparison with other current SOTA reward models.**
>
> Thank you for the suggestion for explaining the baselines more clearly. We have now moved details of baselines from Appendix Section F.6 to the main paper Section 6.1 (utilizing the additional page), explaining the baselines clearly, given their importance. Our main baselines include Vanilla-RM along with robustness-focused ODIN and RRM. It is to be noted that these approaches, including ours, can be applied to any base dataset used for training reward models.
> 1. RM: refers to a standard reward model trained on the base preference dataset using either Pairwise Preference (PairPM) style modelling or Bradley-Terry (BT) style modelling.
> 2. ODIN: ODIN factorizes reward components and disentangles quality and length rewards during training of the reward model.
> 3. RRM (ICLR 2025): refers to a SoTA approach for making RM’s more robust. To the best of our knowledge, this is the best method for robustness while we were writing this paper.
> 4. SoTA Baselines for Neutral Augmentation: In section 6.2, we also compare our Irrelevant Query neutral methodology (IQN) with SoTA neutral augmentation strategies such as paraphrasing (See Crome-PARA, Figure 6), which was recently explored in ReWordBench work (EMNLP’25).
>
> **Results on SoTA datasets**
>
> Since the above approaches can be applied to any base dataset, we also provide results on Skyworks dataset which provides a stronger baseline than UltraFeedback training, presented in Appendix Section C.2.
> We find strong robustness gains and downstream performance on ReWordBench and RewardBench with CROME compared to RRM and other baselines. In particular, CROME improves over RRM in 18/23 ReWordBench transformations,
>
>
> >**Clarity, Improving paper’s writing, experimental design choices could be explained more clearly. Details about RM being trained, at what size (same for the baselines).**
>
> For better clarity we have updated Experimental Settings Section 6.1 as well as all main tables and figures in Section 6 with information of the RM being trained including their size. We also added information about baselines. We clustered some appendix references just before Section 6.2.
>
>
> >**Do you see any benefit in still incorporating more “subjective” preferences in preference tuning? While there might be spurious correlations, certain stylistic preferences could still provide relevant signal for improving an LLMs output distribution AND 460: effect of causal attributes: but should we completely ignore things like length, etc.?**
>
> While subjective and stylistic preferences might perform well on in-distribution (ID) data, if they are spurious for the question, it will make RM’s rely on spurious factors, which will risk poor performance on out-of-distribution data, where the subjective or stylistic preferences do not necessarily align with human preference. Our fixed-rubrics experiment quantitatively sheds light on this, where using non-causal (or subjective, stylistic attributes) resulted in 7.1%, 7.4% and 4.6% degradation on chat-hard, safety and reasoning (See Figure 7). This highlights that in well represented datasets, general subjective preferences cannot be used without harming generalization of reward models.
> Our results on ReWordBench (Figure 4.) also point in this direction, showing that unaligned spurious factors in the prompts or responses lead to out-of-distribution inputs, deteriorating performance of naively trained RMs.
> Having said this, if a particular style or format is important for a query (e.g., if the model has been instructed to output a table-formatted answer), CROME is capable of identifying this (by identifying causal attributes such as instruction-following, formattedness). In this case, the style or format is indeed causal and not spurious.

---

> ### Author Response · Authors · 2025-11-21
> **Author Response - 4**
>
> >**This approach will probably not prevent reward hacking completely? How do you test for that?**
>
> Our framework and causal modelling approach are aimed at preventing reward hacking, and the following assumptions need to hold to completely prevent reward hacking:
>
> i) The train set covers representative examples, ii) Oracle LLM is able to identify the correct causal rubrics, and iii) Augmentations are generated reliably.
>
> In practice, preference datasets are limited, and the oracle may provide some noisy rubrics and counterfactuals; hence, it is not possible to eliminate reward hacking completely. However, we find our approach to work reliably across various upstream and downstream settings for mitigating reward hacking (As seen in Section 6.2 of the main paper).
> However, an important hypothesis emerges from the reviewer's question. As oracle LLMs improve and thereby rubrics and augmentations become more accurate (and closer to theoretical best), the downstream performance of CROME should improve as well. To test this, we have employed an experiment where we compared oracle LLMs: Gemma-3-27B-IT to Gemini-2.0-Flash. When we move from Gemma-3-27B-IT (weaker) to Gemini-2.0-Flash (stronger) as the oracle LLM, robustness and reward modelling performance improve. This experiment is mentioned in Section 6.2 of the main paper under “Robustness to Oracle LLM Choice”. See Table 6 and Appendix Fig. 8 for the results. We include a summary of results below:
>
> **Results on reWordBench and RewardBench**
> |Method|reWordBench ↑|RewardBench ↑|
> |------|-------------|-------------|
> |Vanilla RM|59.97|80.61|
> |RRM|64.68|82.53|
> |**CROME (with Gemma-3-27B-IT as oracle)**|67.90|85.15|
> |**CROME (with Gemini-2.0-Flash as oracle)**|73.07|87.84|
>
>
> >**90: are the causal attributes specified during training?**
>
> No, causal attributes are only used for curating data of the form (q, a, a_c) where a_c is a causal augmentation along a causal attribute c. Once this data is curated (along with neutral augmentations), we do not use this causal attribute (c) during training the CROME reward model.
>
> >**174: don’t understand: what does it mean for a model to be conceptual?**
>
> Our causal model provides a framework for developing a robust reward model; however, implementing its exact realization requires having access to true causal attributes and ideal textual counterfactuals. This is intractable in practice, hence we employ oracle LLMs to produce approximate causal attributes and counterfactuals, while imperfect, they provide the targeted variations that lead to superior performance on downstream robustness tasks. Thus, we call our causal model as “conceptual” to highlight that while the method relies on the existence of a causal model, our implementation is not truly causal.
>
> >**204: how do you check these generations?**
>
> While counterfactual generation can be imperfect, we take concrete measures to reduce the adversarial effect of imperfections. In particular, we perform scalable self-verification + neutral augmentations:
> 1. We perform self-verification where the same oracle LLM is prompted to verify if the causal augmentations were successful in modifying the causal attribute without changing any other attribute. We prompt the LLM to check specifically for changes in the particular causal attribute (See Section K.6 for the prompt). Self-verification led to about 1% data being removed out.
> 2. It is still possible that some spurious attributes also get modified while performing causal augmentations. For this, we use neutral augmentations to provide spurious invariance (Section 3.4)
>
>
> >**270 (now line 285): did you do ablations on whether this is actually necessary?**
>
> Filtering significantly reduces the amount of data to train on, making the training computationally feasible in both our setup as well as RRMs. We exactly follow RRM’s strategy to filter easy responses and did not modify it. Our main aim was to show the benefit our augmentation method brings, which includes the addition of causal augmentation pairs and neutral augmentation pairs. For this, we simply replace RRM’s augmentation strategy with ours. We hence only provide ablations for the augmentation strategies we added (neutral and causal augmentations).

---

> ### Author Response · Authors · 2025-11-21
> **Author Response - 5**
>
> >**Section 6.1: you never mention the size of the RMs that you train, or what they are**
>
> In Section 6.1 first paragraph, we mention that we utilize diverse base LLMs (Gemma-2-9B-IT, Qwen2.5-7B, Gemma-2-2B) and train both Pairwise Preference (PairPM) and Bradley-Terry (BT) style reward models.
> We do this for CROME and all baselines (i.e., all baselines have these as their base model over which training is performed). Hence, the different sizes of RMs we train are 9B, 7B, 2B. We show main results on RewardBench and ReWordBench with these different-sized reward models (Table 3, Figure 4, Appendix C.6 and C.11). Unless otherwise specified, we use the largest reward models (trained over Gemma-2-9B-IT) for alignment results and ablations (Fig. 5-7, Tables 4-7).
> In our updated version, we mention this in Section 6.1 and provide more clarity by updating each of our main tables and figures in Section 6 with the base reward model and RM settings.
>
>
> >**Table 2: main improvements in safety? Why?**
>
> 1. Qualitative analysis shows that oracle LLMs are good at identifying safety as a causal attribute to a harmful or disguised benign question, and upgrades of unsafe answers are often made as refusals. Empirically, this reduces attack success rate (ASR) on harmful prompts in downstream safety-datasets, but does not cause much regression in refusal-to-answer (RTA) on benign prompts (See Figure 5 and Table 15 for these results)
> 2. To empirically see the effect of casual vs neutral augmentations on safety, our neutrals ablations in Figure 12 (safety plot, third from left) show that causal augmentations are the most important for safety improvements. This provides strength to our qualitative analysis in point 1 above.
>
>
>
> >**375 (now line 420): baselines are not explained at all**
>
> We have now moved details of baselines from Appendix Section F.6 to the main paper Section 6.1 (utilizing the additional page). Brief explanation of baselines are also provided as an answer to a question above.
>
> >**Table 5: unclear what is being compared, i.e., what are all the models?**
>
> Here, the goal is to analyse CROME’s performance when Gemma-3-27B-IT is used as oracle, which is a weaker LLM than Gemini-2.0-Flash (our default oracle LLM). We also include a row with our default setting of Gemini-2.0-Flash being used as an oracle. CROME (Gemma-3-27B-IT) and CROME (Gemini-2.0-Flash) refer to our approach when using Gemma-3-27B-IT and Gemini-2.0-Flash as the oracle LLMs, respectively. In this table, the RMs being trained (for baselines and for CROME) start from the Gemma-2-9B-IT model and RMs trained are of pairwise preference (PairPM) style.
> We have now made this more explicit in the text as well
>
> >**465 (now line 508): is that testing whether crome outperforms llm-as-judge?**
>
> Yes, and more specifically, we want to measure if we can simply use the large oracle model (Gemini-2.0-Flash, Gemma-3-27B-IT) as a reward model, given that CROME uses them in its pipeline for generating attributes and augmentations. We find that CROME-trained models significantly outperform the oracles.
>
> >**I could need more detail on data + how much you generate etc.**
>
> (Also See Appendix Section F.4) We generate 5 causal attributes per question, and for each, the oracle generates a causally upgraded and degraded attribute, resulting in 10x the dataset size, and an equal amount of Neutral augmentations. Augmented data and original data is combined. After self-verification and filtering, **we get 3.5x the data size, similar to RRM**.
> Here is a cost breakdown and a budget-matched experiment we performed:
>
> **On Quantifying the cost**:
>
> The cost of inference for our runs is around 50% of the full training cost, as shown below:
>
> - Training cost of RRM is 15 hours of compute, 8×A100s. 20 USD/hr, total training cost is: 20 USD/hr × 15 hr = 300 USD for a standard GCP instance.
> - Inference cost for augmentations for 600k responses at  0.4USD/M output token cost (for Gemini Flash API) costs about 120 USD. This is conservatively < 50% of the training cost of RRM.

---

> ### Author Response · Authors · 2025-11-21
> **Author Response - 6**
>
> (cont. from previous)
>
> **Budget-Matched Experiment:**
>
> We conducted a new budget-matched experiment. We gave the RRM baseline an additional 25%, and 50% of standard preference data, matching CROME's augmentation budget. This could be done since RRM uses only half their augmented data in their recipe. The results below show that this data-boosted RRM still significantly underperformed CROME. This confirms that CROME's structured, causally-guided augmentations are more sample-efficient than simply adding more preference pairs.
> Rewardbench and ReWordBench results:
>
>
> | Model | #Examples | Chat | ChatHard | Safety | Reasoning | Avg-RewardBench | Avg-ReWordBench |
> |-------|-----------|------|-----------|---------|-----------|------------------|------------------|
> | RRM   | X×1.5     | 97.63 | 71.16 | 74.26 | 87.13 | 82.55 | 64.53 |
> | RRM   | X×1.25    | 97.63 | 71.71 | 74.59 | 87.10 | 82.76 | 64.54 |
> | RRM   | X         | 96.93 | 72.04 | 73.78 | 87.36 | 82.53 | 63.92 |
> | **CROME** | **X** | **97.49** | **72.70** | **86.96** | **94.55** | **87.93** | **73.07** |
>
> *Here, X = # original RRM data, 230k examples.*
>
> On ReWordBench, CROME is better than RRM on 21, 20 and 20 out of 23 transformations, for X×1.5, X×1.25 and X number of Examples respectively.
>
> Additionally, we find that when we train CROME with lesser data, equal to X/1.5 AND X/1.25 amounts, we get Avg-RewardBench accuracy to be **85.95** and **85.81** respectively, higher than the original RRM score of 82.53, and Avg-ReWordBench accuracy to be **73.66** and **73.51** respectively, significantly higher than original RRM score of **63.92**, and not much different from the original CROME.
>
>
> >**240 (now line 255): how do you make sure that the irrelevant query is indeed irrelevant?**
>
> Given a large preference dataset and our causally augmented dataset, our limited manual checks gave us the intuition that with high probability, randomizing queries are unlikely to share causal attributes (for large and diverse datasets as in our setting).
>
> For making this more quantitative, we now implemented a filtering process for our Irrelevant Query Neutrals (IQN) using a GPT-4.1-mini Oracle. The Oracle evaluates each randomized query-response pair in the neutrals subset across three dimensions: Causal Linkage, Semantic Alignment, and Functional Contradiction. This assessment uses a $\{-1, 0, 1\}$ Relevance Score: $\mathbf{-1}$ (Completely Irrelevant), $\mathbf{0}$ (Partially Irrelevant), and $\mathbf{1}$ (Perfectly Relevant). A pair of responses $(A_w, A_l)$ is accepted as a valid IQN tie only if both responses meet the condition: $\text{Score}(A_w) \le 0 \land \text{Score}(A_l) \le 0 \land \text{Score}(A_w) == \text{Score}(A_l)$, thus making sure that both responses are equally irrelevant. To ensure the statistical significance of these metrics, we ran the above evaluation 3 times with each run consisting of 100 randomly sampled neutral triples drawn from the full training set.
> |Metrics|Average Proportion|Standard Deviation|Primary Implication|
> |---|---|---|---|
> |are_both_irrelevant_status|99.67%|0.58%|High Irrelevance|
> |are_both_low_score|99.67%|0.58%|Negligible Causal Link|
> |are_scores_equal|99.67%|0.58%|Equal Irrelevance — It's safe to call them neutrals|

---

> ### Author Response · Authors · 2025-11-25
> **Author Response to Reviewer's Question - 1**
>
> Thank you for your prompt response and for spending time in thoroughly reading our responses. We provide our answer to the new questions and some additional experimental results inspired by these questions:
>
> >**Thank you for showing new results on the Ties subset of RewardBench 2. It's good to see that your approach considerably outperforms RRM and RM, both in terms of calibration and accuracy! I have one more question about this evaluation: I see that there models in the RewardBench 2 paper (Table 3), that achieve higher scores on the ties subset. Could you explain why CROME does not outperform them, or why you don't compare against those reward models?**
>
> Since CROME is a model and data agnostic approach, our goal is to compare CROME with baselines in a fair setup where all baselines (used to train a reward model) start from the same base model and preference dataset. We did not include comparisons to the models in Table 3 because they represent reward models trained on different preference datasets, different base models (e.g. LLaMa-3.1-8B), significantly larger models (27B, 70B parameters), or are proprietary models (Gemini, GPT).
>
> CROME (and RRM) are the methodology for curating or augmenting data for robust reward model training, and both can be applied to any fixed combination of {base model, preference dataset, reward modeling setting such as PairPM or Bradley-Terry} for creating robust reward models. In our paper, we choose 2 well established and widely used preference datasets - ultrafeedback and skyworks to apply CROME methodology, and compare with baselines (RM, RRM) that utilize the same datasets. Additionally, for fair comparison we use the same base models for training both baselines, as well as CROME (Gemma2-9B-IT in our previous result on RewardBench2, other base models explored in Appendix Section C.6 ).
>
> Keeping everything else the same, a better reward model can be achieved with a higher quality preference dataset (as is the case for many in Table 3 of RewardBench2 paper). Appendix Section C.2 of our paper shows reward modeling and robustness results using stronger datasets (Skyworks). While in our previous RewardBench2 results in the response above, all models (CROME, RRM, RM) were trained on the widely used UltraFeedback preference dataset, we are happy to extend the RewardBench2 results to a stronger preference dataset below:
>
> We now show results using the **Skyworks preference dataset** (Skywork/Skywork-Reward-Preference-80K-v0.2 on huggingface, also used by other models in Table 3, such a Skywork/Skywork-Reward-Gemma-2-27B model). We evaluated our CROME model (trained with Gemma-2-9B-IT as the base model) trained on randomly sampled 20K datapoints from Skyworks dataset and compared it with RM and RRM trained with the same preference dataset and with the same base model. All models are trained in a pairwise-preference reward modeling setting (PairPM). The higher quality Skyworks dataset leads to stronger overall results (compared to UltraFeedback) for baselines and CROME and we also see some interesting results when comparing with larger models from Table 3 of RewardBench2 paper.
>
> | Model | Overall | Factuality | Precise IF | Math | Safety | Focus | Ties (Accuracy) | Ties (Reward Margin Calibration) | Ties (Weighted Avg) |
> |-|-|-|-|-|-|-|-|-|-|
> | RM | 64.83 | 53.26 | 32.5 | 64.48 | 72.00 | 80.00 | 95.1 | 78.43 | 86.76 |
> | RRM | 68.00 | 54.32 | 36.88 | 66.67 | 75.78 | 89.09 | 92.16 | 78.43 | 85.29 |
> | CROME | 69.44 | 63.37 | 35.00 | 56.83 | 85.11 | 89.09 | 94.12 | 80.39 | 87.25 |
>
> **Findings:**
>
> - **CROME** Achieves a Ties accuracy of **87.25%**, higher than RM and RRM trained with the same base model and preference dataset. This shows that CROME’s gains are agnostic to preference tuning datasets, complementing the reward modeling and robustness results in Appendix Section C.2 and Tables 8, 9  in our paper.
>
> - It is interesting to compare CROME trained on skyworks with some notable models from Table 3 as well.  We see that **CROME outperforms larger baselines on the TIES subset** despite being a smaller 9B model trained on only 20k examples out of the full skyworks training set. **1) vs. Proprietary Models:** CROME trained on skyworks outperforms most proprietary models which are evaluated in llm-as-judge setting, such as google/gemini-2.5-flash (83.4%) and openai/gpt-4o (78.2%) on the TIES subset. **2) vs. 70B Models:** CROME (9B) is competitive with the massive Llama-3.1-70B-Instruct-RM (88.3%), sitting within 1% accuracy. This shows CROME indeed has better calibration compared to various larger models, highlighting the strength of the CROME training methodology. We would like to emphasize that models in Table 3 are trained in various different settings and the fair comparisons can be made only with RRM and RM that are trained in the same settings as CROME (in the table above, in our previous responses, as well as in our paper).

---

> ### Author Response · Authors · 2025-11-25
> **Author Response to Reviewer's Question - 2**
>
> >**Thank you for the data analysis, this is great! I am also satisfied with your answer regarding terminology, subjective preferences, and irrelevant queries! Regarding baselines: Could you clarify what data (and how many data instances) each baseline was trained on?**
>
> For our main results in the paper, we use the Ultrafeedback preference dataset [1] as the base dataset we start from for all baselines and CROME:
>
> **Base RM:** Trained on Ultrafeedback dataset directly,which consists of 60.9K preference pairs.
>
> **ODIN:** The training data for ODIN remains identical to the Base RM and uses 60.9k preference pairs. The key difference is that ODIN architecture includes an auxiliary head for predicting length reward during training. This head is then discarded at inference.
>
> **RRM:** The RRM framework augments data which leads to a 14X data size increase. They then randomly subsample 50% of the data and then filter the dataset using the base reward model (as described in lines 284-286 and lines 1980-1983 in our paper). The filtered subset is combined with the original dataset, which finally gives 230K preference pairs. In our budget matched experiment above (Author Response 6 above, and Appendix Section C.5) we train RRM with 287k and 345K examples by adding back some of the randomly subsampled 50% data (before performing the filtering step). This provided no additional gains as seen in the budget matched experiments.
>
> **CROME:** Our augmentation strategy augments the original dataset by a factor of 20X (10x for causal augmentations, and 10x for irrelevant query neutral augmentations). Following this, we perform filtering (same as RRM, refer to lines 284-286 in the paper) and then combine it with the original dataset. This results in 260k preference pairs. For the budget matched experiments, we train CROME with 153K and 173K examples, which continued to show significantly better performance than RRM (in a budget matched setup when RRM uses 230K examples, as well as compared to RRM trained with 287k and 345K examples).
>
>
> [1] Cui, Ganqu et. al, Ultrafeedback: Boosting language models with scaled ai feedback. ICML 2024
>
>
> Please let us know if you have any other questions or suggestions.

---

> ### Author Response · Authors · 2025-11-28
> **Gentle Follow Up**
>
> We wanted to follow up and thank you for your time and insightful feedback. We are glad to hear that you are satisfied with the various results and analyses we presented, including improved calibration and accuracy results on RewardBench 2.
>
> **We have posted two additional comments to answer your follow-up questions, including 1) Comparing with models in Table 3 of the RewardBench 2 paper and following that comment, comparing with stronger baselines (where we presented results on the Skyworks dataset), and 2) Details of the data and the number of instances each baseline was trained on.**
>
> ---
>
> *We have summarized the new results, analyses, and updates to the paper as part of the general response.*
>
> Should these responses address your concerns, we would greatly appreciate this being reflected through an increase in score. Thank you again for your engagement throughout this process.

---

### Author Response · Authors · 2025-11-28
**Overall Response: Summary of New Results, Analyses and Paper Updates - Part 1**

We sincerely thank the reviewers for providing us with valuable feedback and suggestions, which have helped us strengthen our paper. We are encouraged that reviewers find our work as addressing an important and timely challenge in reward-model robustness (Hbwu, 679r), with a well-motivated causal augmentation framework that is novel and conceptually elegant (679r, Ge7h). Reviewers also highlighted the strength of our synthetic-data methodology leading to more robust and effective reward models (Hbwu, 679r), and appreciated the clarity of the core idea (Ge7h). Finally, we are glad that all reviewers recognized the breadth and rigor of our experiments (Hbwu, 679r, Ge7h), including extensive ablations and evaluations in a variety of settings, which consistently demonstrate CROME’s robustness.

**Feedback from all reviewers is incorporated in the paper (in blue). All concerns have been addressed with new experiments and analyses that further strengthen our work.**

**Summary of Interaction with Reviewer Hbwu:** Rev. Hbwu (on 24th Nov) was satisfied with the strong results on RewardBench2 (RB2) and all other answers in our rebuttal. They further asked for comparisons of CROME with baselines in Table 3 of the RB2 paper and details of the training dataset sizes. For this, we provide a comprehensive comparison of CROME and baselines trained on the Skyworks dataset, further strengthening the RB2 results, and show that CROME outperforms various large and proprietary models in Table 3 of RB2 paper.

**Summary of Response to Reviewer Ge7h:** We provided extensive results and analyses, including human evaluations of rubric quality and evaluating rubric consistency across models. These address all of Reviewer Ge7h's concerns.

----

### **Major New Experiments, Analyses and Corresponding  Updates to the Paper:**

>**1. Results on RewardBench2 and performance on the TIES subset**

We test our trained models directly on RewardBench 2. When using Ultrafeedback as the preference dataset and Gemma-2-9B-IT as the base model to train reward models, We find that CROME outperforms RRM and RM on the average RewardBench2 score (overall) by approximately 1.5% and 5.5%, respectively, and improves on the TIES subset by approximately 2% and 4% over RRM and RM, respectively, signifying improved calibration. When trained on the Skyworks dataset, we find CROME achieves a Ties accuracy of 87.25%, higher than RM and RRM. Interestingly, CROME outperforms various larger baselines (including most proprietary models) listed in Table 3 of the RewardBench2 paper.

**These results have been added to the end of Section 6.2 and in Figure 8.**

>**2. Quantifying of the Cost of Augmentations and Budget-Matched Experiments**

We provided reviewers with details of the cost of augmentations (presented in Section C.5 of the original paper). Following this, we did a new budget-matched experiment by providing RRM with additional examples, or CROME with fewer examples to train on, leading to equal compute being utilized by the two methods. When RRM is trained with larger number of examples to match the budget (1.5 times the examples), we find CROME to provide similar significant increase in RewardBench accuracy of around 5.5%, and CROME being better than RRM on 21, out of 23 reWordBench transformations (9% improvement on average). We continue to see similar significant gains when CROME is trained with 1.5 times lesser data and compared with the original RRM in a budget matched setting.

**These results have been added to Appendix Section C.5**

>**3. Consistency of rubrics across oracle choices and Human Evaluation of Quality of Rubrics**

We performed a quantitative analysis of similarity of rubrics obtained from 2 oracles on examples from the ultrafeedback dataset and measured their similarity using 2 frontier models (Gemini-3-pro and GPT-5.1). We find high overlap (>70%) between rubrics obtained from 2 strong oracles used in our work.

For further analysis of the quality of rubrics, we compare the alignment between rubrics obtained from Gemini 2.0 Flash, and compare it with those obtained via Humans, in a controlled setup where we have a fixed set of rubrics. This small-scale study demonstrated strong alignment between ranking of rubrics between human and the model.

**Results have been added to Section N and Table 16 of the Appendix**

>**4. Analyses of the properties of generated synthetic data**

We provide a detailed analysis of the synthetically generated data along three dimensions,
- Analyzing length distributions of original and causally upgraded and degraded responses, along with how the distribution of the difference of lengths between chosen and rejected responses changes with the data we generate.
- Analysis of sycophancy of generated synthetic responses.
- Comparative analysis of refusal rates for safety-critical prompts for base preference dataset and the causal upgraded response.

**Analyses have been added to Section O of the Appendix**

---

> ### Author Response · Authors · 2025-11-28
> **Overall Response: Summary of New Results, Analyses and Paper Updates - Part 2**
>
> ### **Rewrites, Expositional Details and Corresponding Updates to the Paper:**
>
> >**1. Clarity and Mathematical Notations**
>
> As suggested by reviewer 679r, we **revisit Sections 3 and 4** of the paper to add more natural prose and explanations along with mathematical notation, to improve readability. In section 3.2, we have explained that there might be exogenous factors which are common causes of $SP(A)$ and $C(A),Q$ in text, with the heavier causal notations moving to the appendix. In section 3.3, we ensured to introduce and explain each variable. We have also clarified how we do preference annotation on the answer pairs in our synthetically generated dataset.
>
> **Changes have been added to Section 3 and Section 4 of the Main Paper**
>
> >**2. Details of Baselines, Experimental settings, Model details**
>
> While all details of baselines, experimental settings and model details were part of the main paper or appendix, we utilized the additional page to include some of these details from the appendix to the main paper. As a result, Section 6 contains more details of baselines, experimental settings and model type and size details. All tables contain detailed experimental settings making them self-contained.
>
> **Changes have been added to Section 6 of the Main Paper**
>
>
> -----
>
> *Please find answers to specific questions and feedback in individual reviewer responses below.
> Through a variety of additional experiments and analyses, we have addressed all weaknesses raised by the reviewers. We thank them for their time and service in reviewing our work. Should there be any other pending suggestion or feedback, we would be happy to incorporate it and engage in further discussion.*

---

### Meta-Review · Area_Chair_vDvp · 2026-01-06

**Summary:**

The paper presents CROME, a framework for training robust reward models (RMs) by using an oracle LLM to identify causal rubrics (the true drivers of quality) and generating targeted synthetic data. The method addresses reward hacking by creating pairs that enforce sensitivity to causal attributes and invariance to spurious ones (like length or formatting).

Initially, the reviewers had different attitudes. Reviewer Hbwu expressed significant concerns regarding the lack of rigorous data analysis, missing baselines, and the potential for the term causal to be an overstatement. Reviewer Ge7h appreciated the novelty of the idea but questioned the quality/consistency of the oracle rubrics and the computational cost. Reviewer 679r was strongly positive in general, praising the experimental design.

The consensus shifted toward Acceptance following the rebuttal. The authors provided significant new evidence, including performance on RewardBench2, a Ties subset evaluation, human validation of rubrics, and budget-matched experiments proving the sample efficiency of their approach.

**Reviewer Concerns:**

Addressed concerns:
- Authors provided new results on RewardBench2. CROME significantly outperformed baselines in the Ties subset (improving calibration by ~3%), showing it effectively differentiates between correct-vs-correct and correct-vs-incorrect
- Authors conducted a quantitative analysis on length bias, sycophancy, and refusal rates. They demonstrated that while upgraded responses are sometimes longer, the model does not reward verbosity alone if it lacks causal quality
- Authors provided a human study (MRR of 0.93) showing high alignment between model-generated rubrics and human-identified ones. They also showed ~70-75% overlap in rubrics between different oracle models (Gemma vs. Gemini)
- Authors quantified the cost (inference is <50% of training cost) and performed a budget-matched experiment showing that simply giving baselines more data does not close the performance gap

Outstanding concerns:
- Reviewer Hbwu noted that causal and counterfactual might be slightly overselling since the interventions are LLM-approximated rather than formally causal. The authors addressed this by labeling their model as conceptual and adding caveats, but the philosophical distinction remains a minor point of contention
- While the method is effective, its performance is inherently capped by the quality of the frontier model (Oracle) used to generate rubrics.

**Reviewer Scores:**

For reviewer Hbwu, most concerns were addressed, although there are some minor concerns about the concept usage. As reviewer Hbwu expressed satisfied for the reply about termonology, data analysis and new baseline results, the score tends to change from 2 to 4. For the remaining reviewers, most concerns are resolved and their positive score will maintain unchanged.

---

### Decision · Program_Chairs · 2026-01-26

Accept (Poster)